# Development of Machine Learning Model for Prediction of Demolition Waste Generation Rate of Buildings in Redevelopment Areas

**DOI:** 10.3390/ijerph20010107

**Published:** 2022-12-21

**Authors:** Gi-Wook Cha, Se-Hyu Choi, Won-Hwa Hong, Choon-Wook Park

**Affiliations:** 1School of Science and Technology Acceleration Engineering, Kyungpook National University, Daegu 41566, Republic of Korea; 2School of Architectural, Civil, Environmental and Energy Engineering, Kyungpook National University, Daegu 41566, Republic of Korea; 3Industry Academic Cooperation Foundation, Kyungpook National University, Daegu 41566, Republic of Korea

**Keywords:** waste management, demolition waste, machine learning, optimal predictive model, waste generation rate, redevelopment area

## Abstract

Owing to a rapid increase in waste, waste management has become essential, for which waste generation (WG) information has been effectively utilized. Various studies have recently focused on the development of reliable predictive models by applying artificial intelligence to the construction and prediction of WG information. In this study, research was conducted on the development of machine learning (ML) models for predicting the demolition waste generation rate (DWGR) of buildings in redevelopment areas in South Korea. Various ML algorithms (i.e., artificial neural network (ANN), K-nearest neighbors (KNN), linear regression (LR), random forest (RF), and support vector machine (SVM)) were applied to the development of an optimal predictive model, and the main hyper parameters (HPs) for each algorithm were optimized. The results suggest that ANN-ReLu (coefficient of determination (R^2^) 0.900, the ratio of percent deviation (RPD) 3.16), SVM-polynomial (R^2^ 0.889, RPD 3.00), and ANN-logistic (R^2^ 0.883, RPD 2.92) are the best ML models for predicting the DWGR. They showed average errors of 7.3%, 7.4%, and 7.5%, respectively, compared to the average observed values, confirming the accurate predictive performance, and in the uncertainty analysis, the d-factor of the models appeared less than 1, showing that the presented models are reliable. Through a comparison with ML algorithms and HPs applied in previous related studies, the results herein also showed that the selection of various ML algorithms and HPs is important in developing optimal ML models for WG management.

## 1. Introduction

The World Bank reported in 2018 that municipal solid waste (MSW) will increase to 3.4 billion tons per year by 2050 [1]. Construction and demolition waste (CDW), which represents 35–40% of the total waste generated [2], is rapidly increasing, and 70–90% of the CDW is demolition waste (DW) [3,4]. Although the CDW is a problem in itself owing to the enormous amount produced, it also increases social and environmental costs by harmfully affecting the environment. An increase in the amount of waste not only deteriorates urban environments, but it also causes various environmental and health risks, such as groundwater contamination, land degradation, increased cancer risks, child mortality, and congenital anomalies in newborn babies [5]. Governments and researchers in related industries have, therefore, focused on waste management (WM) and have introduced cutting-edge technologies and intelligent systems.

Information on waste generation (WG) is a prerequisite and useful tool for various WM strategies, such as landfill space planning, the calculation of charges to pollution sources or polluters, the calculation of subsidies for recycling, and the establishment of WM policies by related companies [6]. As such, many researchers have conducted studies on a WG prediction for WM and have actively utilized artificial intelligence (AI) technologies in recent years. To develop WG prediction models for WM, researchers have applied numerous machine learning (ML) algorithms in various data environments (e.g., various key factors for model development, the dataset size, and the types of such input variables as categorical and numerical variables that constitute a dataset). Owing to its excellent ability to model complex mechanisms, ML has been successfully applied in the development of predictive models for the management of MSW or CDW [7,8].

According to Abdallah et al. [9] and Xia et al. [10], artificial neural networks (ANNs), support vector machines (SVMs), and linear regression (LR) have been widely used as ML algorithms for WG prediction. Some researchers have applied algorithms such as k-nearest neighbor (KNN), decision tree (DT), and random forest (RF) toward the development of WG prediction models. For example, Cha et al. [11] and Song et al. [12] applied an ANN for CDW prediction, whereas Golbaz et al. [13], Liang et al. [14], and Soni et al. [15] applied it toward the development of MSW prediction models.

Cai et al. [16], Cha et al. [11], and Song et al. [12] used an SVM toward the development of CDW generation prediction models. Abbasi et al. [17,18], Abbasi and Hanandeh [19], Abunama et al. [20], and Golbaz et al. [13] used an SVM to predict MSW generation, whereas Kumar et al. [21] used it to predict the generation rate of plastic waste.

Along with ANN and SVM algorithms, LR has also been frequently used for WG prediction by many researchers. Azadi and Karimi-Jashni [22], Chhay et al. [23], Fu et al. [24], Golbaz et al. [13], Kumar and Samadder [25], Montecinos et al. [26], and Wei et al. [27] applied an LR algorithm to MSW prediction models, whereas Wu et al. [28] applied it to CDW prediction models.

In addition to ANN, SV, and LR algorithms, DT-, RF-, and KNN-based approaches have also been used by researchers in WG predictions. Cha et al. [29] developed 11 types of DW generation prediction models using a DT algorithm. Kannangara et al. [30] and Rosecký et al. [31] applied a DT algorithm to the prediction of generated MSW. Cha et al. [32,33] used RF for the development of CDW generation prediction models, whereas Dissanayaka and Vasanthap-riyan [34], Kumar et al. [21], and Nguyen et al. [35] utilized RF for the development of MSW prediction models. In addition, Abbasi and Hanandeh [19] and Nguyen et al. [35] used a KNN algorithm for WG prediction.

The performance results of the AI models developed using ML algorithms in the above previous studies for WG prediction vary depending on the characteristics of the data (e.g., data size and input variable type) and data processing method applied. In addition, the various studies showed different performance results depending on the selection of the hyper parameter (HP) values. Therefore, for the predictive performance of AI models applying ML algorithms, it is important to develop optimized models that consider the algorithm type, data characteristics (e.g., dataset size and type of input variables), and HPs applied. Based on this, various ML algorithms must be applied and optimal HPs must be selected when developing WG prediction models for specific data.

The main purpose of this study is to develop optimal ML predictive models for predicting the DW generation rate (DWGR). More specifically, ML predictive models suitable for the characteristics of the data used in this study must be developed, and optimal ML predictive models for predicting the DWGR must be proposed by deriving appropriate modeling methods and HPs for various ML algorithms. To achieve these purposes, the following research was conducted.

The DWGR data were collected from 160 buildings, and a dataset was constructed by applying data preprocessing for the raw data.ANN, SVM, LR, KNN, and RF were considered ML algorithms for the development of the DWGR prediction models.The DWGR prediction models were developed by deriving the optimal HPs for each ML algorithm.The leave one out cross-validation (LOOCV) technique was used for model validation. The mean squared error (*MSE*), root mean square error (*RMSE*), coefficient of determination (*R*^2^), and mean absolute error (*MAE*) were used as the statistical metrics for evaluating the model performance.Optimal ML models for the DWGR prediction were proposed by evaluating the performance of the developed models.

The rest of this paper is organized as follows. In Section 2, the materials and methods of the study are described. In Section 3, the performance results of the DWGR prediction models developed in this study are analyzed. In Section 4, the results of previous related research are discussed along with the models developed in the present study. In Section 5, key findings are summarized, and the limitations of this study and future research directions are discussed.

## 2. Materials and Methods

In this section, the data and data processing method used for the DWGR prediction, the characteristics of the applied algorithms, and a validation and evaluation of the predictive models are described. Descriptions of the collected data and the data preprocessing method, including the categorical variables used, are provided in Section 2.1. An introduction to and the application of the supervised ML algorithms used in this study are described in Section 2.2. HP tuning for securing the optimal performance of the ML models is detailed in Section 2.3. Finally, the validation and evaluation of the ML models developed for the DWGR prediction are described in Section 2.4. Figure 1 shows the research flow of the present study.

### 2.1. Data Collection and Preprocessing

The DWGR data used in this study was the DW discharge (kg/m^−2^) records collected from demolition sites in redevelopment areas in two Korean cities (i.e., Daegu and Busan). For the collection of data on DWGR, information on the building features (i.e., location, structure, usage, wall type, roof type, gross floor area (GFA), and number of floors) was collected through a direct survey of 160 buildings prior to building demolition. Table 1 and Figure 2 show the building status and statistical analysis results based on the collected data. As can be seen in Table 1, the DWGR characteristics exhibited clear differences in mean values depending on the location, usage, structure, wall type, and roof type. Because the redevelopment areas mainly consisted of low-rise buildings, most of the DWGR data was distributed within the range of 1800 (kg·m^−2^) or lower. The scatter plot in Figure 2 shows the DWGR distribution characteristics according to the building area. Overall, the buildings in project B area showed large GFA and DWGR values, and the number of floors was found to be large. Many reinforced and concrete-brick structure buildings were distributed within the project B area.

The building features of the dataset used in this study, such as the location, structure, usage, GFA, number of floors, wall type, and roof type, were the main factors affecting the DWGR. In this study, seven building features (i.e., location, structure, usage, GFA, number of floors, wall type, and roof type) were used as input variables for the DWGR prediction. Therefore, the correlations between the DWGR and the seven building features can be expressed as Equation (1) below. In addition, the DWGR is defined as Equation (2) in this study.
(1)$$DWGR=f\;\left(location,\;structure,\;usage,\;GFA,\;number\;of\;floors,\;wall\;type,\;roof\;type\right)\;$$
(2)$$DWGR_{i}=\frac{\sum A\;of\;building_{i}\;}{GFA\;of\;building_{i}}$$
where $DWGR_{i}$ is the demolition waste generation rate (kg·m^−2^), *A* is the waste generation (kg), and GFA is the gross floor area (m^2^), all of building i.

Data pre-processing was applied to the collected dataset to improve the performance of the ML models. The data pre-processing used in this study includes label encoding for converting categorical input variables into numerical variables and the normalization process for achieving a uniform data scale. Data normalization was conducted using Equation (3).
(3)$$x_{normalization}=\frac{x-x_{min}}{x_{max}-x_{min}}$$
where x is the data element, $x_{max}$ is the maximum number of data, and $x_{min}$ is the minimum number of data.

### 2.2. Applied Machine Learning Algorithms

ANN, SVM, and LR algorithms were most frequently used to develop AI models for WG prediction in the field of CDW and MSW [9], and various studies showed high prediction performance results. In addition, DT, RF, and KNN algorithms have also been applied as ML algorithms for WG prediction by various researchers [19,29,30,31,32,33,34,35] in the field of CDW and MSW and have shown excellent predictive performance results. Therefore, this study preferentially considered ANN, SVM, LR, DT, RF, and KNN algorithms, and this study considered RF rather than DT. This is because RF compared to DT, which is a single algorithm, can avoid overfitting and can expect superior predictive performance [32,33,34,35,36].

#### 2.2.1. Artificial Neural Network

An ANN is a computing system composed of multiple layers and neurons. Such a network is classified into feedforward and feedback neural networks, the former of which have been widely used in the field of waste management because they are relatively simpler than the latter and achieve a superior performance [37]. The basic structure of an ANN consists of three layers (i.e., input, hidden, and output) and nonlinear transfer functions that make it possible to learn the nonlinear and linear relationships between the input and output neurons constituting several layers. An ANN is one of the most frequently used algorithms for AI model development in the field of WM [9].

#### 2.2.2. Support Vector Machine

The basic idea of an SVM is to map the input data onto the feature space through a nonlinear map, and linear decision functions are created in the feature space. In this instance, the aim of an SVM is to find the optimal decision function. The SVM then nonlinearly maps a linear decision function of the feature space onto the original space through kernels [38]. An SVM model is intended to alleviate the main drawback of parametric regression, which is a novel powerful learning machine based on statistical learning theory [39]. SVM models exhibit a good problem solving performance with small samples, nonlinearity, and high-dimensional features [40]. SVMs have been utilized as a main algorithm for AI models in various fields along with ANNs, and they have been applied to WG prediction by many different researchers.

#### 2.2.3. Linear Regression

An LR model is a linear equation composed of output values for specific input values based on a statistical ML algorithm. Although LR allows an easy interpretation and low computation cost, it is prone to biased results [35]. Nevertheless, LR is a significantly attractive model because it can allow the results to be easily interpreted using a simple basic algorithm [9,35]. Owing to these benefits, LR has been consistently used as an ML algorithm for the development of WG prediction models.

#### 2.2.4. K-Nearest Neighbors

KNN has been widely applied to regression and classification problems in various fields owing to its simplicity and intuitiveness. In general, KNN is more suitable for low-dimensional data with a small number of input variables [35]. KNN is based on a clustering algorithm with supervised learning, which is based on a distance calculation using training data and a pre-defined k value, to find the data nearest to the k value [41]. KNN is considered more suitable for low-dimensional data than data with many input variables [35]. Whereas various ML algorithms have been applied to WG prediction, KNN has been used in only two studies [19,35].

#### 2.2.5. Random Forest

An RF is one of the most powerful ML algorithms. Proposed by Breiman [36], it is a representative ensemble technique based on bagging for the creation of bootstrap sampling. An RF extracts multiple subsets (bootstrap sampling) from the original dataset and creates a tree (a.k.a. a weak learner) for each subset. For final result prediction, a strong learner is determined through a majority vote from the result of each tree. Through this process, an RF can avoid an overfitting and is less affected by outliers as the number of trees increases. An RF exhibits a higher predictive performance than other ML algorithms even when the classes are imbalanced [36]. RFs have recently been used by researchers for WG prediction in the fields of construction, demolition, and solid waste management and have exhibited a superior predictive performance compared to other ML algorithms [11,33,35].

### 2.3. Hyper-Parameter Tuning

The HPs of a model significantly affect the predictive performance, robustness, and generalization ability of the model. Therefore, in this study, the HPs of the algorithms applied in this study (i.e., ANN, KNN, LR, RF, and SVM) were tuned to derive optimal predictive performance models. To derive an optimal performance, HPs such as the activation function, number of hidden layers, number of neurons, regularization, and number of iterations were tuned for the ANN model. For the KNN model, the HPs were tested for distance metrics (i.e., Euclidean, Manhattan, and Chebyshev) and the k value (a.k.a. k-neighbors). For the LR model, three types of regularization methods (i.e., ridge, lasso, and elastic) were considered, and the regulation strength (alpha value), such as L1 and L2, was tuned. For the RF model, the number of trees indicated by an n_estimator was first considered [42], and the tree depth and branch classification criteria indicated by max_depth and min-samples_leaf, respectively, were tuned. For the SVM model, four kernel types (i.e., linear, polynomial, radial basis function (RBF), and sigmoid) were considered, and the optimal model was derived by tuning such HPs as the cost, epsilon, and iteration. In this study, the ML models were developed using combinations of optimal HPs, as shown in Table 2.

### 2.4. Model Validation and Evaluation

#### 2.4.1. Model Validation

LOOCV is a special type of a *k*-fold cross-validation and is considered a suitable validation method when the sample size is small [43,44]. Therefore, LOOCV is used to evaluate the performance of an algorithm when there are few instances in the dataset [45]. Compared to a 10- or k-fold cross-validation, LOOCV can obtain stabler results when small datasets are targeted because it uses all samples as the testing and training data to secure sufficient training and validation sets [11,32,33,46,47]. Therefore, in this study, LOOCV was applied as a model validation method when considering the size of the dataset.

MAE (Equation (4)), RMSE (Equation (5)), R^2^ (Equation (6)), and MSE (Equation (7)) were used to evaluate the performance of the DWGR prediction models developed in this study. A satisfied model generally yields a high R^2^ value and low MAE, MSE, and RMSE values.
(4)$$MAE=\frac{\sum_{i=1}^{n}\left|y_{i}-x_{i}\right|}{n}$$
(5)$$RMSE=\sqrt{\sum_{i=1}^{n}\frac{{\left(y_{i}-x_{i}\right)}^{2}}{n}}$$
(6)$$R^{2}=1-\frac{\sum_{i=1}^{n}{\left(y_{i}-x_{i}\right)}^{2}}{\sum_{i=1}^{n}{\left(y_{i}-{\bar{x}}_{i}\right)}^{2}}$$
(7)$$MSE=\sum_{i=1}^{n}\frac{{\left(y_{i}-x_{i}\right)}^{2}}{n}$$
where $x_{i}$ is the observed value of the quantity of the generated DW, $y_{i}$ is the predicted quantity of the generated DW, ${\bar{x}}_{i}$ is the average observed quantity of the generated DW, and *n* is the number of samples.

The performance of the ML model should be verified through a multi-criteria process, without making false exaggerations or compromising the accuracy of the model [48,49]. Therefore, the ratio of percent deviation (RPD, Equation (8)) [49] was calculated to evaluate the performance of the DWGR prediction models.
(8)$$RPD=\frac{Standard\;deviation}{RMSE}$$

Depending on the value of RPD, the performance of the model is classified as follows. An RPD < 1.0 indicates very poor model/predictions, and their use is not recommended; an RPD between 1.0 and 1.4 indicates poor model/predictions where only high and low values are distinguishable; an RPD between 1.4 and 1.8 indicates fair model/predictions, which may be used for assessment and correlation; an RPD values between 1.8 and 2.0 indicates good model/predictions where quantitative predictions are possible; an RPD between 2.0 and 2.5 indicates very good, quantitative model/ predictions; and an RPD > 2.5 indicates excellent model/predictions [49].

#### 2.4.2. Model Uncertainty Analysis

This study calculated a 95% prediction uncertainty (95 PPU) to judge the uncertainty of the final model. To calculate the 95 PPU, the bounds are determined by the probability of the cumulative distribution of experimental samples for values of 97.5% (*X_U_*) and 2.5% (*X_L_*) for 1000 outputs. The higher the proportion of observations within the 95 PPU range, the smaller the uncertainty and vice versa. The 95 percent prediction uncertainties (95 PPU) are calculated as [50,51,52]:(9)$$\mathrm{Bracketed}\;\mathrm{by}\;95\;\mathrm{PPU}=\frac{1}{n}count\;\left(\left.N\right|X_{L}\leq N\leq \;X_{U}\right)\times 100$$
where *n* indicates the number of the observed data. *N* increases with observations falling between the corresponding *X_L_* and *X_U_* increase; the “Bracketed by 95 PPU” denotes the number of observed data bracketed by a 95% confidence interval, and this value is equal to 100 when all of the observed data are within the range of *X_L_* ≤ *N* ≤ *X_U_* [50,51,52,53,54,55].

The goodness of fit of the model is evaluated by the d-factor, the average distance between the lower 95 PPU (2.5th, *X_U_*), and the upper 95 PPU (97.5th, *X_L_*), which can be evaluated as [51,54]:(10)$$d-factor=\frac{{\bar{d}}_{x}}{{\bar{\sigma }}_{x}}$$
(11)$${\bar{d}}_{x}=\frac{1}{n}\;\sum_{i=1}^{n}\left(X_{L}-X_{U}\right)$$
where *n* is the number of observed data points, ${\bar{d}}_{x}$ is the average distance between the upper (97.5th) and lower (2.5th) bands, and ${\bar{\sigma }}_{x}$ is the standard deviation of measured variable *X*. A value of less than 1 is a desirable measure for the *d-factor* [54].

## 3. Results and Discussion

### 3.1. Performance Results

Figure 3 and Table 3 show the results of the MSE, RMSE, MAE, R^2^, and RPD performance evaluations of the ML models developed for the DWGR prediction. The ANN-ReLu model exhibited the highest DWGR prediction performance, and the SVM-polynomial and ANN-Logistic models also showed excellent predictive results with an R^2^ value of 0.88 or higher. Considering that the MAE values of the ANN-ReLu, SVM-polynomial, and ANN-Logistic models were 71.93, 72.75, and 74.01, respectively, the errors from the average of the observed values (987.18 kg·m^−2^) were 7.3%, 7.4%, and 7.5%, respectively, indicating highly accurate predictive performances. The RF and LR models also showed good performance results with an R^2^ value of 0.8 or higher, and according to the results of the RPD values (Table 3), the models such as ANN-ReLu, SVM-Polynoimal, ANN-Logistic, RF, ANN-tanh, and SVM-RBF show excellent DWGR prediction performance. The KNN model, however, did not exhibit good performance results compared to the other algorithms, with an R^2^ value ranging from 0.631 to 0.776. In addition, the RPD values of KNN models are between 1.4 and 1.8, so the predictive performance of the KNN models is fair. A similar result was also found by Abbasi and Hanandeh [19], whose KNN model for MSW prediction showed a low R^2^ value of 0.51.

Although the models developed in this study showed different results depending on the algorithm applied, significantly different performance results were also observed depending on the selection of HPs despite the application of the same algorithm (Figure 3, Table 3). For the ANN model, the ANN-ReLu model (MSE of 10,160.03, MAE of 71.93, RMSE of 100.80, R^2^ of 0.900, and RPD of 3.16) exhibited a higher predictive performance than the ANN-Logistic, ANN-tanh, and ANN-Identity models, and its performance was found to be significantly different from that of the ANN-Identity model (MSE of 17,850.72, MAE of 97.59, RMSE of 133.61, R^2^ of 0.824, and RPD of 2.38). For the SVM model, SVM-polynomial (MSE of 11,265.73, MAE of 72.75, RMSE of 106.14, R^2^ of 0.889, and RPD of 3.00) also showed a higher predictive performance than the SVM-RBF, SVM-Linear, and SVM-Sigmoid models, and it exhibited a significant difference in performance from SVM-Sigmoid (MSE of 39,240.29, MAE of 159.07, RMSE of 198.09, R^2^ of 0.613, and RPD of 1.61). This confirms that the selection of HPs is quite important for ML model development depending on the algorithm type and that it has a significant influence on the predictive performance of the ML model.

### 3.2. Comparison of Prediction Results and Uncertainty Analysis of Best Models

Among the models developed in this study, the ANN-ReLu, SVM-polynomial, and ANN-Logistic models exhibited the highest predictive performances, and they can, therefore, be considered the best models for the DWGR prediction. Figure 4 shows the correlations between the observed values and the values predicted by the ANN-ReLu, SVM-polynomial, and ANN-Logistic models. The values predicted by these models had errors fewer than ±20% of the observed values in most cases. It appears that the predicted values of the SVM-polynomial model showed more stable results than those of the ANN-ReLu and ANN-Logistic models with an error rate of within ±20% of the observed values. The predicted values of the ANN-ReLu and ANN-Logistic models, however, appear to be more concentrated along the line with a correlation of 1 than those of the SVM-polynomial model. Figure 5 shows the observed values and those values predicted by the ANN-ReLu, SVM-polynomial, and ANN-Logistic models. The predicted values of the models simulate the observed values well, indicating the excellent DWGR prediction performance of the ANN-ReLu, SVM-polynomial, and ANN-Logistic models. The average observed value was 987.18 (kg·m^−2^), whereas the average predicted values of the ANN-ReLu, SVM-polynomial, and ANN-Logistic models were 987.06 (kg·m^−2^), 988.09 (kg·m^−2^), and 990.65 (kg·m^−2^), respectively.

In this study, 95 PPU and d-factor were applied as the uncertainty analysis index of the developed models. A favourable model is attained when the value of the d-factor is small and the proportion of data observed within 95 PPU is high. Figure 6 shows the uncertainty indices (95 PPU and d-factor) results of three different models (i.e., ANN-ReLu, SVM-polynomial, and ANN-Logistic models). The number of observed data lying within the 95 PPU shows the lowest result in the ANN-ReLu model (71%), and the SVM-polynomial and ANN-Logistic models show the same result at 76%. And the d-factors of ANN-ReLu, SVM-polynomial, and ANN-Logistic models are 0.43, 0.44, and 0.43, respectively, all the d-factor values showing results less than 1. Considering the 95PPU and d-factor results, the ANN-ReLu, SVM-polynomial, and ANN-Logistic models developed for the DWGR prediction are judged to be sufficiently superior models.

### 3.3. Importance of Input Variables

Figure 7 shows the relative importance of the input variables used in the ANN-ReLu, SVM-polynomial, and ANN-Logistic models developed for the DWGR prediction. For the ANN-Relu model, the location and structure showed a relatively higher influence than the other input variables. For the SVM-polynomial model, the location, floor area, structure, and wall type exhibited a high relative importance as input variables. In the case of the ANN-Logistic model, the floor area, number of floors, and wall type were relatively important, along with the location and structure. The location and structure were commonly used as input variables with a high relative importance for the ANN-ReLu, SVM-polynomial, and ANN-Logistic models. By contrast, the usage and roof type showed a low relative importance as input variables for all three models. In addition, it can be confirmed through Figure 7a,c that the importance of the input variables varies depending on the HPs applied despite the application of the same ANN algorithm.

## 4. Discussion and Recommendations

This study attempted to apply various ML algorithms for the DWGR prediction, and the DWGR prediction models were developed using five categorical (i.e., location, usage, structure, wall type, and roof type) and two numerical (i.e., floor area and number of floors) input variables. The top-two models showing the optimal performance in this study were the ANN-ReLu (R^2^ of 0.900) and SVM-polynomial (R^2^ of 0.889) models. The ANN model exhibited an optimized performance with the ReLu activation function and 100 neurons. The SVM model was optimized using the polynomial kernel, regression cost = 100, and ε = 30 as the HPs (refer to Table 2 for the remaining parameters). To predict the MSW and CDW generation rates, the authors of previous studies [13,17,19,21] also attempted to develop ML models with an optimal performance using various algorithms, the research results of which varied depending on the sizes of various datasets, input variables, and HPs used. For example, Kumar et al. [21] developed models by applying ANN, SVM, and RF algorithms to predict the rate of plastic waste generation. To develop optimal performance models, the authors applied a logistic activation function and one hidden layer with five neurons for the ANN model and a linear kernel parameter for the SVM model. In addition, the RF model was optimized when the number of trees, the total number of prediction nodes at each decision tree, and the smallest sized terminal node were 100, 5, and 5, respectively. They found that the ANN (R^2^ = 0.75) and SVM (R^2^ = 0.74) models were superior to the RF model (R^2^ = 0.66). Golbaz et al. [13] developed ANN, SVM, and hybrid ANN and SVM models to predict hospital solid waste (HSW) generation. They found that the SVM model achieved the highest total HSW prediction performance with an R^2^ value of 0.90. Information on the kernel parameter of the SVM model used remains unconfirmed. In addition, the ANN model was optimized in a structure with ten neurons in one hidden layer, and the R^2^ of the ANN model in this instance was 0.78 (validation). Abbasi et al. [17] tested the predictive performance of different models according to the linear, polynomial, RBF, and sigmoid kernels of the SVM algorithm for MSW prediction in the cities of Tehran and Mashhad. As the research results indicate, the RBF kernel exhibited the highest predictive performance (R^2^ of 0.702 and 0.756 in Tehran and Mashhad, respectively). However, the polynomial kernel achieved the best results among the SVM models.

As examined above, the algorithm types and the HPs of the models developed in both this study and previous research were optimized at various values. This is because the dataset size and input variables differ for each study. Therefore, by applying various algorithms and HPs, it is necessary to develop ML models with the optimal performance for WG prediction. In this study, optimal performing DWGR prediction models were developed by testing various algorithms, such as ANN, KNN, LR, RF, and SVM, and by applying the HPs to the developed models. Considering the dataset environment of this study (i.e., DWGR data from 160 buildings, 5 categorical input variables, and 2 numerical input variables), the ANN and SVM algorithms were deemed appropriate for the DWGR prediction. Therefore, to predict the DWGR or MSW for datasets with similar characteristics as the input variables used in this study, researchers or stakeholders in related fields can attempt to develop ML models for WM by referring to the results of this study. For the development of ML models, however, it is necessary to find models with optimal performance results by testing them with various algorithms and HPs. In addition, in this study, the DWGR prediction models were developed using single algorithms. In recent years, however, studies have focused on improving the performance of single algorithms by developing various hybrid ML models [11,14,15,16,55]. Therefore, based on the results of this study, there is also room for improvement in the proposed model, and further research will be required to improve the predictive performance.

## 5. Conclusions

In this study, various machine learning (ML) algorithms (i.e., ANN, KNN, LR, RF, and SVM) were applied to predict the DWGR of buildings in redevelopment areas in Korea, and hyper parameters (HPs) were optimized to improve the performance of the DWGR prediction models. ANN-ReLu (R^2^ = 0.900, RPD = 3.16), SVM-polynomial (R^2^ = 0.889, RPD = 3.00), and ANN-Logistic (R^2^ = 0.883, RPD = 2.92) models were suggested based on a high predictive performance with an R^2^ value of 0.88 or higher. And the RPD values of the presented models all showed excellent predictive performance results higher than 2.5. From the average observed values (987.18 kg·m^−2^), these models had error rates of 7.3%, 7.4%, and 7.5%, respectively, indicating a high predictive performance. In addition, in the uncertainty analysis results, all three models presented in this study ensured sufficient reliability because the d-factor values were lower than 1.

When the models developed in this study were compared with those from previous studies developed for the prediction of MSW and CDW generation, the values of the optimized HPs were significantly different despite applying the same algorithm. This shows that the selection of the algorithm and HPs is important in developing ML predictive models for WG management and that for optimal ML model development, it is necessary to optimize the models by applying various algorithms and testing the HPs, which is due to the size of the dataset used and the different input variables applied to each study.

The DWGR models suggested in this study (i.e., ANN-ReLu, SVM-polynoimal, and ANN-logistic) exhibited excellent performance results as predictive models with an R^2^ value ranging from 0.883 to 0.900. However, further studies are needed to improve the uncertainties from the results of 95 PPU. Considering that various attempts have recently been made to improve the performance of ML predictive models by combining single ML algorithms and developing hybrid models, further research is required to improve the performance of the DWGR prediction models developed in this study.

## Figures and Tables

**Figure 1 ijerph-20-00107-f001:**
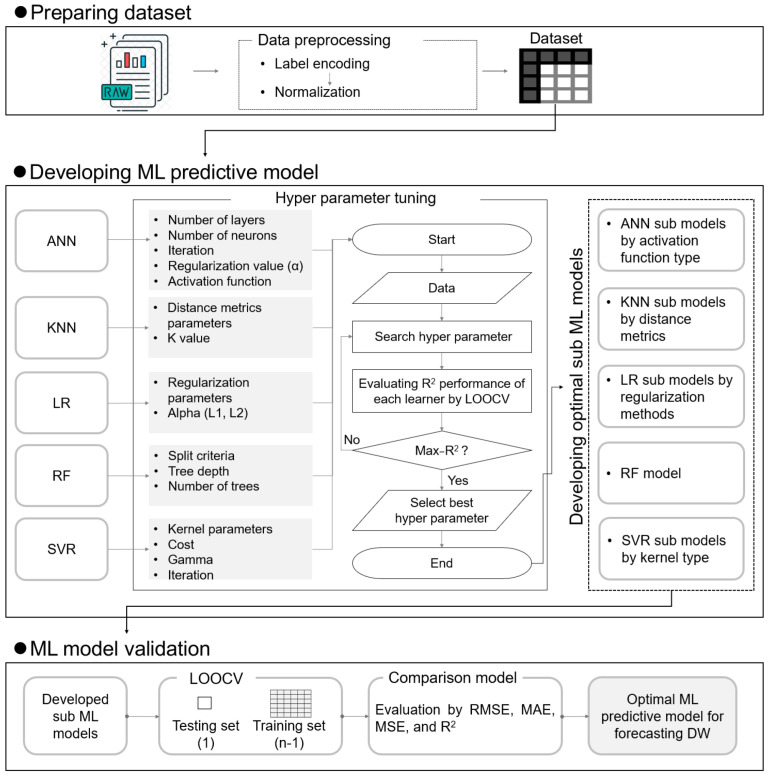
Research flow for development of DWGR prediction model.

**Figure 2 ijerph-20-00107-f002:**
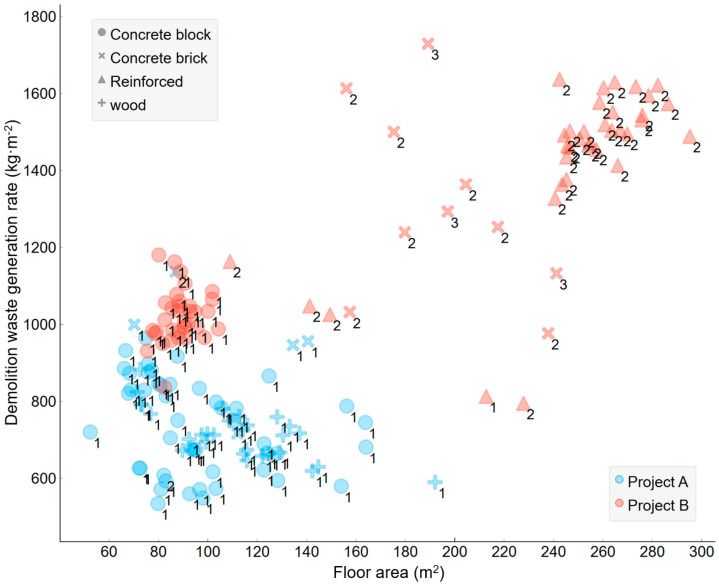
Scatter plot showing the distribution between the building characteristics and the DWGR (numeric labels indicate the number of floors).

**Figure 3 ijerph-20-00107-f003:**
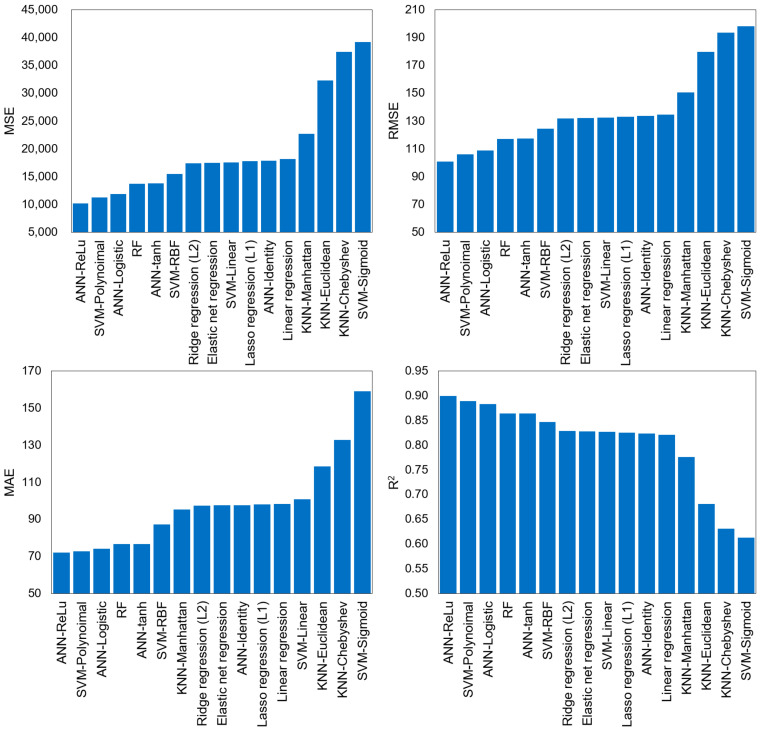
Performance results of ML models developed for DWGR prediction.

**Figure 4 ijerph-20-00107-f004:**
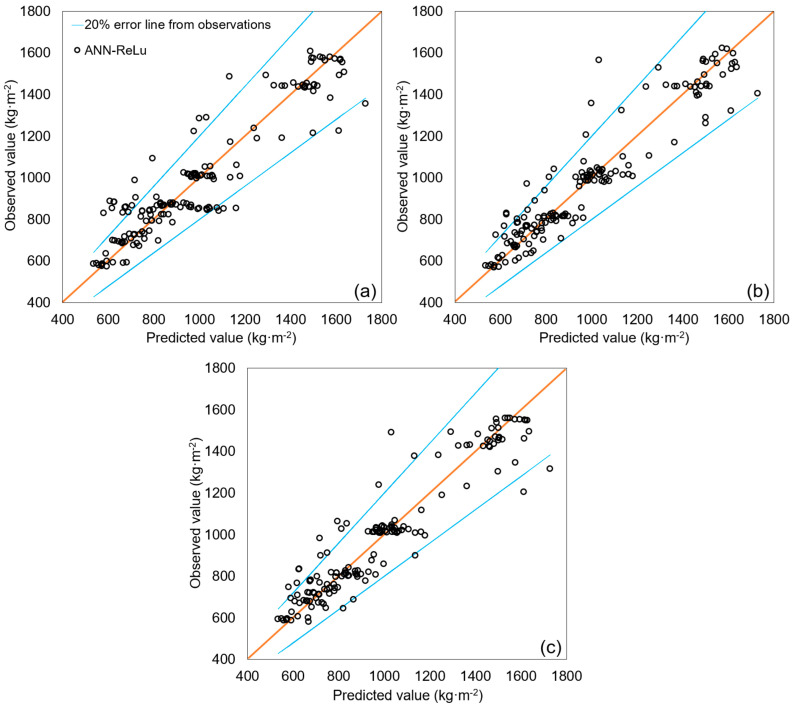
Comparison of the correlation between observed and predicted values: (**a**) ANN-ReLu, (**b**) SVM-polynomial, and (**c**) ANN-Logistic (The blue line in the graph means ±20% error from the observed value).

**Figure 5 ijerph-20-00107-f005:**
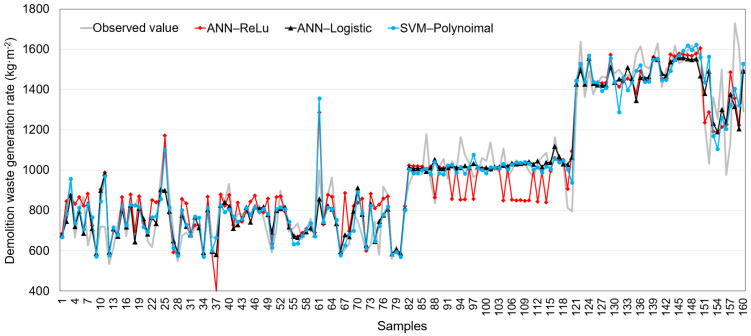
Comparison of observed and predicted values by ANN-ReLu, SVM-polynomial, and ANN-Logistic.

**Figure 6 ijerph-20-00107-f006:**
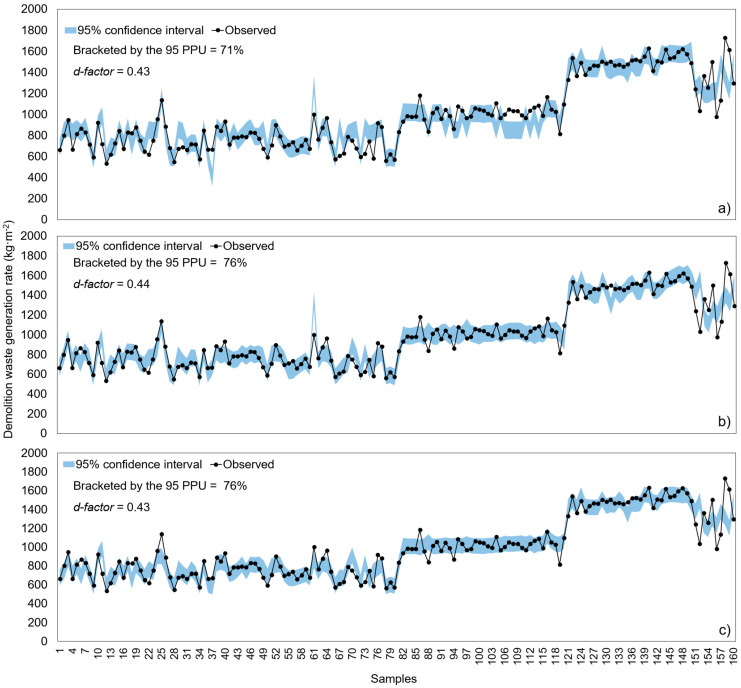
Observed and 95% confidence intervals for estimates of DWGR by (**a**) ANN-ReLu, (**b**) SVM-polynomial, and (**c**) ANN-Logistic.

**Figure 7 ijerph-20-00107-f007:**
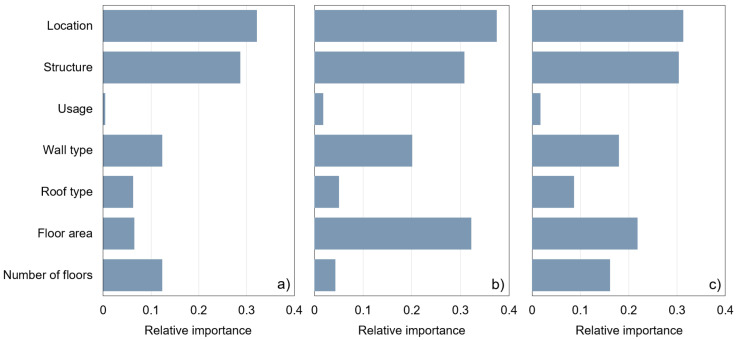
Relative importance of input variables by (**a**) ANN-ReLu, (**b**) SVM-Polynomial, and (**c**) ANN-Logistic models.

**Table 1 ijerph-20-00107-t001:** Building status and statistical analysis of dataset used in this study.

Category		Number of Buildings	DWGR (kg·m^−2^)				GFA (m^2^)
			Total	Min	Mean	Max	
Location	Project A	81	60,072.6	534.8	741.6	1137.3	8301.0
	Project B	79	97,876.3	795.2	1238.9	1729.1	13,627.3
Usage	Residential	135	126,624.9	534.8	938.0	1629.4	16,994.4
	Residential & commercial (Re/Co)	25	31,324.0	750.9	1253.0	1729.1	4933.8
Structure	Reinforced Concrete	35	50,062.4	795.2	1430.4	1637.0	8652.1
	Concrete block (Con_blo)	81	69,225.6	534.8	854.6	1180.5	7539.8
	Concrete brick (Con_bri)	15	21,782.2	717.0	1452.1	1729.1	2500.4
	Wood	29	21,905.8	590.5	755.4	883.0	3236.0
Wall type	Brick	32	31,870.0	590.5	995.9	1729.1	4556.0
	Block	121	121,096.4	534.8	1000.8	1637.0	16,579.0
	Soil	7	4982.5	668.0	711.8	759.8	4556.0
Roof type	Slab	37	45,448.0	717.0	1228.3	1729.1	7003.5
	Slab roofing tile (Slab/R.t)	33	40,035.1	931.0	1213.2	1637.0	5080.8
	Slab & slate	3	3930.9	813.5	1310.3	1614.8	719.6
	Slate	13	7796.1	534.8	599.7	681.8	1384.9
	Roofing tile (R.t)	74	60,738.8	580.0	820.8	1576.5	7739.5

**Table 2 ijerph-20-00107-t002:** Hyperparameters considered for the development of machine learning based predictive models.

Machine Learning Algorithms	Hyper Parameters			
	Title		Tested Values	Selected
ANN	Identity	number of neurons	Range (10, 200, step = 10)	10
		iteration	Range (10, 1000, step = 10)	50
		regularization	Range (0.001, 1000)	1000
	Logistic	number of neurons	Range (10, 200, step = 10)	40
		iteration	Range (10, 1000, step = 10)	400
		regularization	Range (0.001, 1000)	20
	Relu	number of neurons	Range (10, 200, step = 10)	140
		iteration	Range (10, 1000, step = 10)	200
		regularization	Range (0.001, 1000)	600
	Tanh	number of neurons	Range (10, 200, step = 10)	90
		iteration	Range (10, 1000, step = 10)	1000
		regularization	Range (0.001, 1000)	60
KNN	Euclidean	k_neighbors	Range (1, 30, step = 1)	4
	Manhattan			4
	Chebyshev			24
LR	Ridge	alpha	Range (0.0001, 1000)	2
	Lasso			1
	Elastic			0.6
RF	n_estimators		Range (10, 100, step = 10), Range (100, 200, step = 20), and Range (250, 500, step = 50)	30
	max_depth		Range (1, 15, step = 1)	6
	min_samples_split		2, 3, 4, 5, 6, 7, 8, 9, 10	7
	max_features		1, 2, 3, 4, 5, 6	6
SVM	Linear	Cost	Range (0.1, 500)	5
		eplison	Range (0.1, 500)	100
		iteration	Range (10, 500, step = 10)	50
	Polynomial	Cost	Range (0.1, 500)	500
		eplison	Range (0.1, 500)	30
		iteration	Range (10, 500, step = 10)	100
	Rbf	Cost	Range (0.1, 500)	500
		eplison	Range (0.1, 500)	0.1
		iteration	Range (10, 500, step = 10)	90
	Sigmoid	Cost	Range (0.1, 500)	50
		eplison	Range (0.1, 500)	0.5
		iteration	Range (10, 500, step = 10)	80

**Table 3 ijerph-20-00107-t003:** The ratio of percent deviation (RPD) performance of ML models developed for DWGR prediction.

ML Models	RPD Value	Performance
ANN-ReLu	3.16	Excellent
SVM-Polynoimal	3.00	Excellent
ANN-Logistic	2.92	Excellent
RF	2.72	Excellent
ANN-tanh	2.71	Excellent
SVM-RBF	2.56	Excellent
Ridge regression (L2)	2.42	Good
Elastic net regression	2.41	Good
SVM-Linear	2.40	Good
Lasso regression (L1)	2.39	Good
ANN-Identity	2.38	Good
Linear regression	2.36	Good
KNN-Manhattan	2.11	Good
KNN-Euclidean	1.77	Fair
KNN-Chebyshev	1.65	Fair
SVM-Sigmoid	1.61	Fair

## Data Availability

All data included in this study are available upon request by contact with the corresponding author.

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
