# Peer review of "Development of Machine Learning Model for Prediction of Demolition Waste Generation Rate of Buildings in Redevelopment Areas"

_ijerph, 2022, doi:10.3390/ijerph20010107_

Round 1

Reviewer 1 Report

The study deals with “Development of machine learning model for prediction of demolition waste generation rate of buildings in redevelopment areas. Overall the study is good. But the following comments should be addressed before it can be accepted.
1) Why the authors have particularly chosen RF, ANN, K-nearest and SVM for predicting the demolition waste?
2) The authors may include the uncertainty analysis for emphasizing more on the accuracy of the developed models.
3) The literature review about the developed ML based models must be improved by incorporating its latest application in the field of engineering (e.g., https://www.sciencedirect.com/science/article/pii/S0266114421000480 ; TRGEO (36), September 2022, 100827, and Neural Computing and Applications v 33, pages14861–14885 (2021)
4) Improve the Fig. 1 visibility (pixels).
5) Kindly provide the weights and biases of the developed ANN models within manuscript. This is important as without it, it would not be possible to replicate the results for the future researchers.
6) The accuracy of the models are only judged on four criteria; however Multicriteria approach such as OBJ function and external validation approach should be utilized to judge the accuracy to the models.
7) Line 289: The values predicted by these models had errors in fewer than 20% of the observed values in most cases. These error lines are drawn at ± 20%, kindly correct it.

Author Response

Thank you for your review of the completeness of this paper. The revision of this paper reflecting your review is as follows:

Point 1: Why the authors have particularly chosen RF, ANN, K-nearest and SVM for predicting the demolition waste?

Response 1: I am grateful for your advice for improving the completeness of this paper. Reflecting your opinion, in 2.2 Applied machine learning algorithms, the reasons for selecting the ML algorithms applied in this study were described. Please refer to Lines 166 to 174.

Point 2: The authors may include the uncertainty analysis for emphasizing more on the accuracy of the developed models.

Response 2: I sincerely appreciate that your opinions have been of great help in improving the quality of the thesis. In reflection of your opinion, the information on uncertainty analysis has been added and is as follows.

1) 2.4.2. In model uncertainty analysis (lines 279 to 298), information on uncertainty analysis methods has been added.

2) The title of Section 3.2 was changed to 'Comparison of Prediction Results and uncertainty analysis of Best Models' (line 337), and information on uncertainty analysis (lines 355 to 365) was added.

3) Figure 6 showing uncertainty results has been added.

4) Uncertainty-related information was also supplemented in the conclusion and abstract.

Point 3: The literature review about the developed ML based models must be improved by incorporating its latest application in the field of engineering (e.g., https://www.sciencedirect.com/science/article/pii/S0266114421000480; TRGEO (36), September 2022, 100827, and Neural Computing and Applications v 33, pages14861–14885 (2021)

Response 3:Thanks for your comments. Added related references. Please refer to Line 582 to 596.

Point 4: Improve the Fig. 1 visibility (pixels).

Response 4: Thanks for your comments. The visibility of Figure 1 has been improved. And other pictures also improved visibility.

Point 5: Kindly provide the weights and biases of the developed ANN models within manuscript. This is important as without it, it would not be possible to replicate the results for the future researchers.

Response 5: Thank you for your review. We respect your opinion regarding the ANN model. I also checked the related paper you linked. As you pointed out, disclosing the weights and biases of the ANN may be necessary for reproducibility of the study results. However, I would like to mention some points that need to be considered for this paper.

We are trying to find an algorithm suitable for the used data among different ML algorithms applied in this study, and presenting a model with the best predictive performance. We gave priority to ML algorithms that are frequently used in construction, demolition, and municipal solid waste-related fields, and here we find an appropriate algorithm and develop a model. In this process, methods and results must be equally provided for all applied algorithms, and results biased to a specific algorithm are not considered appropriate. In addition, such results deviate from the intention to be shown in this paper, and at the same time, it seems that problems may arise in the composition of the paper. So please consider our opinion on this.

Point 6: The accuracy of the models are only judged on four criteria; however Multicriteria approach such as OBJ function and external validation approach should be utilized to judge the accuracy to the models.

Response 6: I appreciate your comments to improve the quality of the thesis. Based on your comments, the ratio of percent deviation (RPD) has been added as a Multicriteria approach. Please refer to lines 264 to 278 for information on RPD. And the results of the added RPD were reinforced in the abstract, results, and conclusions sections.

Point 7: Line 289: The values predicted by these models had errors in fewer than 20% of the observed values in most cases. These error lines are drawn at ± 20%, kindly correct it.

Response 7: Thanks for the precise point. That part has been corrected. Please refer to lines 343 and 345.

Reviewer 2 Report

SUMMARY

In this study, an investigation has been done on the development of machine learning models for predicting the demolition waste generation rate of buildings. Thus, Machine Learning (ML) algorithms were applied to the development of an optimal predictive model, optimizing the main hyper parameters (HPs).

FINDINGS

ANN-ReLu (R2 0.900), SVM-polynomial (R2 0.889), and ANN-logistic (R2 0.883) are the best ML models for predicting DWGR.

By comparing with ML algorithms and HPs applied in previous related studies, it is also shown that selecting various ML algorithms and HPs is important when developing optimal ML models for WG management.

STRENGTHS

The article is structured in a clear and concise way. The introduction gives a correct background, and the figures, tables and mathematical expressions facilitate the reading of the paper.

WEAKNESSES

The paper is well organized and correctly addresses the subject of study. Only one minor change would be necessary for the acceptance of the manuscript. Thus, presentation of the article could be improved:

-          The font size of figures 1, 3, 4, 5 and 6 could be increased.

-          Equations 4 and 5 do not appear.

Author Response

Thank you for your review of the completeness of this paper. The revision of this paper reflecting your review is as follows:

Point 1: The paper is well organized and correctly addresses the subject of study. Only one minor change would be necessary for the acceptance of the manuscript. Thus, presentation of the article could be improved:

- The font size of figures 1, 3, 4, 5 and 6 could be increased.

- Equations 4 and 5 do not appear.

Response 1: Thank you for your comments on improving the quality of this manuscript.

- According to your advice, the font size of all pictures has been adjusted. Please check Figures 1 to 7.

- There was a mistake in the numbering of expressions. The equation numbers were checked again and corrected.

Round 2

Reviewer 1 Report

The authors have made good effort to address the comments. 

The referencing 50 to 54 seems abundent and can easily be compacted by more relavent and compact referencing e.g., https://doi.org/10.1016/j.trgeo.2022.100783 and/or https://link.springer.com/article/10.1007/s00500-021-06628-x